# Current State and Challenges in Developing Respiratory Syncytial Virus Vaccines

**DOI:** 10.3390/vaccines8040672

**Published:** 2020-11-11

**Authors:** Carlotta Biagi, Arianna Dondi, Sara Scarpini, Alessandro Rocca, Silvia Vandini, Giulia Poletti, Marcello Lanari

**Affiliations:** 1Pediatric Emergency Unit, Department of Medical and Surgical Sciences (DIMEC), Sant’Orsola University Hospital, 40138 Bologna, Italy; carlotta.biagi@aosp.bo.it (C.B.); sscarpini@gmail.com (S.S.); alessandrorocca2003@yahoo.it (A.R.); giuliapoletti89@gmail.com (G.P.); marcello.lanari@unibo.it (M.L.); 2Pediatric and Neonatology Unit, Imola Hospital, 40026 Imola (Bologna), Italy; s.vandini@ausl.imola.bo.it

**Keywords:** RSV, vaccine, prevention

## Abstract

Respiratory syncytial virus (RSV) is the main cause of acute respiratory tract infections in infants and it also induces significant disease in the elderly. The clinical course may be severe, especially in high-risk populations (infants and elderly), with a large number of deaths in developing countries and of intensive care hospitalizations worldwide. To date, prevention strategies against RSV infection is based on hygienic measures and passive immunization with humanized monoclonal antibodies, limited to selected high-risk children due to their high costs. The development of a safe and effective vaccine is a global health need and an important objective of research in this field. A growing number of RSV vaccine candidates in different formats (particle-based vaccines, vector-based vaccines, subunit vaccines and live-attenuated vaccines) are being developed and are now at different stages, many of them already being in the clinical stage. While waiting for commercially available safe and effective vaccines, immune prophylaxis in selected groups of high-risk populations is still mandatory. This review summarizes the state-of-the-art of the RSV vaccine research and its implications for clinical practice, focusing on the characteristics of the vaccines that reached the clinical stage of development.

## 1. Introduction

### 1.1. Epidemiology

Respiratory syncytial virus (RSV) is often responsible for severe seasonal respiratory disease in infants and elderly people, and it causes a great burden on health systems worldwide. RSV clinical presentation in children involves respiratory infections ranging from mild upper to severe lower respiratory tract infections (LRTI), including pneumonia or bronchiolitis [1,2,3,4]; in fact, it was estimated that RSV may cause up to 22% of all severe acute LRTI in young children. A recent systematic review [3] reported that worldwide in 2015, RSV caused 33.1 million episodes of LRTI, resulting in nearly 3.2 million hospitalizations and 59,600 in-hospital deaths in children younger than 5 years. In infants younger than 6 months, RSV caused 1.4 million hospital admissions and 27,300 in-hospital deaths [3]. The estimated overall RSV mortality in 2015 was 118,200 deaths in infants younger than 5 years. Moreover, RSV is responsible for pulmonary morbidity and hospitalizations also in the elderly and high-risk adults [5], causing more than 17,000 deaths for underlying pneumonia and circulatory complications [6]. Risk factors for severe RSV infections in paediatric populations that may require hospitalization and Intensive Care Unit admission include prematurity, chronic lung disease of prematurity or bronchopulmonary dysplasia, hemodynamically significant congenital heart disease, congenital or acquired immunodeficiency and severe neuromuscular disease.

### 1.2. Molecular Characteristics of Respiratory Syncytial Virus

RSV is an enveloped, non-segmented, negative-sense, single-stranded RNA virus of the *Orthopneumovirus* genus and *Pneumoviridae* family. The envelope of RSV includes four proteins associated with the lipid double layer: the matrix (M) protein, the small hydrophobic (SH) protein and the two glycosylated surface proteins: F (Fusion) and G (attachment glycoprotein). The proteins, which are directly involved in infectivity and development of the respiratory disease, are F and G: the G protein determines the attachment of the virus to host epithelial cells, while the F protein is involved in the entry of the virus, through the fusion of viral and cellular membranes, as well as the subsequent insertion of the viral RNA into the host cell; the F protein is also responsible for fusion of infected cells with adjacent cells, resulting in the formation of the characteristic syncytia [7]. Moreover, both F and G proteins induce the neutralizing antibody immune response by the host [8]. Three types of epitopes have been identified in the G protein: (1) conserved epitopes, detectable in all viral strains; (2) group-specific epitopes, expressed only by the same antigenic group; and (3) strain-specific epitopes, which are present only in specific strains of the same antigenic group and expressed in the C-terminal hypervariable region of the G protein ectodomain [9]. The F glycoprotein is derived from an inactive precursor containing three hydrophobic peptides: (1) the N-terminal signal peptide, mediating translocation of the nascent polypeptide into the lumen of the endoplasmic reticulum; (2) the transmembrane region near the C-terminus, which links the F protein to the cell and viral membranes; and (3) the so-called fusion peptide, which inserts into the target cell membrane and determines the fusion process. The binding of the prefusion F protein to the cell surface is followed by its activation and structural changes, which determine the fusion of the membrane of the virus with the airway epithelial cells of the host and lead to the formation of syncytia. There are two major subgroups of RSV, A and B, usually both detectable during the same epidemic season, even if temporal and geographic clustering may occur [10,11]. RSV infections with Group A have a higher incidence and higher transmissibility than Group B [12]. The antigenic variability between the two groups is determined by differences in the G glycoprotein (35% homology between the G glycoprotein of strains A and B) [13]. For this reason, many antibodies targeting the G protein may be subtype specific, while antibodies against the F protein have a neutralizing effect both against RSV A and B. Vaccine candidates against RSV use different antigenic targets [14]. Most vaccines in clinical trials use the F protein, because it is highly conserved and facilitates viral fusion with host cells. Current vaccine candidates use pre-F and post-F as vaccine antigens. Site zero (∅) of the pre-F protein has been recently discovered and is one of the major antigenic targets [15]. Other less frequent vaccine antigens, used alone or in combination with others, include the RSV envelope associated glycoprotein G and the SH protein, as well as the internal proteins: nucleocapsid (N), M and M2-1. Besides the F protein, the G protein is the only other target for neutralizing antibodies on the viral surface.

### 1.3. Prophylaxis

Prevention of RSV infection is based on primary and secondary prophylaxis. Preventive measures are critical because an etiologic treatment against RSV is not available. Primary prophylaxis is essentially based on hygienic measures aimed to prevent the diffusion of respiratory virus infections (hand washing, use of face masks in case of symptoms, disinfections of objects and surfaces) [16]. Secondary prophylaxis is based on the administration of monoclonal antibodies (mAbs) to high-risk patients during the epidemic season. The only commercially available mAb is palivizumab, while more recent ones are being studied in ongoing clinical trials [17,18,19]. No vaccine is currently available for active immunization against RSV, even if several are a matter of ongoing clinical trials [14]. One of the hardest challenges in the development of a safe and effective vaccine is enhanced respiratory disease (ERD), a side-effect that occurred in the 1960s subsequently to administration of formalin-inactivated RSV vaccines [20,21]. Moreover, it is still matter of concern if a vaccine determines an absolute protection against RSV infections; encouraging data demonstrate that cell-mediated immunity, mucosal IgA and serum neutralizing antibodies are inversely related to disease severity [14].

### 1.4. Target Population

Considering the epidemiology of RSV, preventive measures are particularly useful for target populations. Currently, passive immunization with mAbs is reserved for high-risk infants [17], as detailed in national and international guidelines. Risk factors considered in the guidelines for prophylaxis include severe prematurity, congenital heart diseases, bronchopulmonary dysplasia, severe respiratory and neuromuscular diseases. The clinical trials of vaccines against RSV are run on paediatric populations, old adults or pregnant women with the aim to prevent RSV infections in high-risk populations (Table 1). RSV was also observed to affect long-term respiratory morbidity since young infants with RSV LRTI subsequently had an increased risk for asthma and recurrent wheezing [22]. For this reason, the development of a safe and effective vaccine may be useful to prevent the onset of respiratory morbidity if extended to cohorts of naive infants during the first months of life. Since severe RSV infections often involve newborns and very young infants, the strategy of vaccination during pregnancy may be useful to determine an effective immunity at early ages when active immunization is not feasible, as maternal antibodies are transferred efficiently through the placenta. Vaccines in clinical development are grouped into four categories: particle-based, vector-based, subunit and live-attenuated or chimeric vaccines. In the following paragraphs we discuss in detail the different categories of vaccines and the state-of-the-art of the clinical trials.

## 2. Particle-Based Vaccines

Particle-based vaccines are synthesized by self-assembling nanoscopic particles that expose multiple copies of a selected antigen on their surface and mimic the native virions [23]. Thanks to the high copy number of the selected antigen and the immune-boosting properties of the particle matrix, these vaccines elicit strong humoral and cellular immune responses [23]. Moreover, the lack of the viral genome required for replication make them safe. To date, two nanoparticle-based RSV vaccines have been tested in clinical trials: the RSV F nanoparticle vaccine, without or with an aluminium adjuvant, and SynGEM (Table 2). The RSV F nanoparticle vaccine, developed by Novavax, is composed of recombinant F-proteins, which have the post-F morphology and are formulated with polysorbare 80 [24]. The conformation of the F proteins is a singly cleaved pre-fusogenic form [25,26]. This vaccine candidate is being evaluated in women of childbearing age, pre-school children (2–6 years old) and the elderly (≥60 years old). In Phase I clinical trials, it has proven to be well tolerated and highly immunogenic in all the target populations [27,28,29,30]. Subsequent clinical trials have been conducted in pregnant women and the elderly. ResVax is a maternal RSV F nanoparticle vaccine with an aluminium phosphate adjuvant. It is being developed to protect infants from RSV disease via maternal immunization. ResVax has been shown to be safe in a Phase II trial enrolling 50 healthy third-trimester pregnant women and to elicit RSV neutralizing antibodies and palivizumab-competing antibodies that are efficiently transferred to the infants [31]. This maternal vaccine has been the object of the PREPARE trial, a Phase III multi-country, randomized, placebo-controlled trial evaluating the vaccine efficacy against RSV-LRTI in infants from birth to 90–180 days of life [32]. From December 2015 to March 2019, 4363 pregnant women with expected delivery near the beginning of the RSV season were enrolled in the study. Women were randomized to receive a single intramuscular dose of the vaccine (120 μg RSV-F protein adsorbed to 0.4 mg aluminium) or a placebo between 28 and 36 weeks of pregnancy. ResVax failed to meet the primary outcome of prevention of medically significant LRTI. However, it showed 44% efficacy in reducing RSV-LRTI hospitalization. Moreover, it demonstrated 39.4% efficacy in reducing RSV-specific medically significant LRTI and 58.8% efficacy in reducing RSV-related severe hypoxemia in young infants (<3 months of age). In addition, pneumonia was 49.4% less common in infants of the vaccinees than the placebo recipients [32]. According to these results, ResVax is the first RSV vaccine to show efficacy in a Phase III clinical trials, even if it did not meet the desired primary outcome. Because of the failure to achieve the primary outcome, according to the Food and Drug Administration (FDA) and the European Medicines Agency (EMA) recommendations, Novavax will conduct an additional Phase III clinical trial to confirm the efficacy of ResVax [33]. Regarding older adults, the non-adjuvanted RSV F nanoparticle vaccine failed to demonstrate efficacy in a 2015 Phase III clinical trial (RESOLVE trial). This study enrolled 11,850 adults ≥60 years of age randomized to receive 135 µg of the vaccine via intramuscular injection or placebo. The vaccine did not reduce the incidence of RSV-positive moderate to severe LRTI, nor the incidence of all RSV-symptomatic respiratory diseases [34]. However, the vaccine was associated with a 61% reduction in hospitalizations due to exacerbations of chronic obstructive pulmonary disease (COPD). On the basis of this result, Novavax is planning to start an efficacy trial to evaluate the COPD exacerbations as a prospective endpoint [35]. Following the unsatisfactory results of the RESOLVE trial, in 2017, Novavax conducted a second Phase II clinical trial in the elderly. The aim was to evaluate the safety and immunogenicity of single- or two-dose regimens of the RSV F vaccine with and without adjuvants (aluminium phosphate or Matrix-M1). All formulations and regimens were well tolerated. The study showed that both adjuvants increased the magnitude, duration and quality of the immune response versus the non-adjuvanted RSV F vaccine [35]. These data support the inclusion of the RSV F vaccine adjuvant formulations in future elderly trials. Another nanoparticle-based RSV vaccine that has been tested in clinical trials is SynGEM, developed by Mucosis. SynGEM is a mucosal vaccine containing the RSV F protein in the prefusion conformation bound to a bacterium-like particle (BLP) derived from *Lactococcus lactis*. BLP has the role to present the vaccine antigen in a more natural conformation and to boost the immune system of the virus [36]. The safety and tolerability of SynGEM have been evaluated in a Phase I trial enrolling 48 healthy volunteers [37]. The participants were randomly assigned to receive intranasal SynGEM (low or high dose) or a placebo according to a prime-boost schedule with the boost vaccination at 28 days after the prime. SynGEM was generally well tolerated and induce persistent antibody responses in healthy adults. High-dose recipients (350 mg RSV-F protein, 5 mg BLP) achieved plateau responses in antibody titres after the first dose. SynGEM induced bursts of plasmablast activity and mucosal IgA and elicit systemic RSV-specific antibodies (non-neutralizing and palivizumab-like antibodies) for at least 6 months. However, no detectable RSV neutralizing antibodies (F protein site ∅-specific antibodies) were found, despite preclinical data that had shown protection [38]. Future studies are needed to optimize the immunogenicity of SynGEM. In 2017, Mucosis declared bankruptcy, thus any further development of the vaccine may be planned by other companies. The SynGEM technology platform is now made available by the private company Virtutax [39].

Particle-based vaccines are promising both for young infants through immunization of the pregnant mothers and for the elderly. ResVax has already reached Phase III clinical trials for pregnant women, and hopefully new trials will soon allow its marketing approval. On the other hand, for elderly people, the use of RSV F vaccine adjuvant formulations in future trials may increase the magnitude, duration and quality of the induced immune response.

## 3. Vector-Based Vaccines

Vector-based vaccines use a carrier vector to deliver RSV antigens and induce an immune response against RSV components exploiting the adjuvant effect of the vector. Due to the chimeric nature of the vectors, there is no risk of reversion to wild-type RSV and of ERD. However, the presence of pre-existing anti-vector immunity or its potential development may challenge the clinical use of these vaccines. To date, eight RSV vector-based vaccines have been tested in clinical trials (Table 3). The MVA-BN-RSV vaccine has been developed by Bavarian Nordic and it is based on a non-replicating modified vaccinia Ankara (MVA) virus, previously used as a vaccine against smallpox [43,44]. This vaccine candidate displays the RSV surface proteins F and G (for both A and B subtypes) and two internal RSV proteins N and M2. A Phase I study has shown that it can induce broad cellular and humoral immune responses against RSV [45]. A subsequent Phase II trial, evaluating the safety and immunogenicity of the vaccine in 420 older adults (≥55 years), showed that it is well tolerated and lead to a persistent immune response for at least 6 months with the possibility to boost it at 12 months [46]. On the basis of these results, a Phase III trial in the elderly has been planned for initiation in 2021 [47]. ReiThera developed two novel, recombinant, viral-vectored vaccine candidates for RSV, PanAd3-RSV and MVA-RSV. Both vaccines use RSV F, N and M2 proteins as antigens, delivered, respectively, by a replication-defective simian Adenovirus (PanAd3) and MVA vector. PanAd3-RSV and MVA-RSV, given in different combinations and by different routes of administration (PanAd3 either intramuscularly or intranasally, MVA intramuscularly), were well tolerated and immunogenic in healthy adults despite pre-existing natural immunity to RSV [48]. Similar results occurred in the elderly, who presented both humoral and cellular vaccine responses in a Phase I study [49]. The development of these vaccines has been halted since 2015. Three other virus-vectored vaccines use an Adenovirus vector to deliver RSV antigens. VXA-RSV-f, developed by Vaxart, is a molecularly adjuvanted Adenovirus serotype 5 based RSV vaccine encoding the F protein. It is being developed for older adults. Oral delivery of this vaccine has been shown to induce a strong humoral response against RSV in cotton rats [50]. Following these pre-clinical results, VXA-RSV-f was tested orally in healthy adults in a Phase I trial. The study ended in September 2017, but the results have not yet been published [51]. Janssen developed Ad26.RSV.Pre-F vaccine for the elderly and children, based on the human Adenovirus strain 26 vector expressing the F protein stabilized in the pre-fusion conformation [52]. This vaccine candidate demonstrated superior efficacy to the previously tested Ad26. RSV.FA2 that displayed post-F as an antigen. A Phase II study evaluated the co-administration of a seasonal influenza vaccine with the Ad26.RSV.preF vaccine in 180 healthy elderly people, showing an acceptable safety profile and no evidence of interference in immune response [53]. Two other Phase II trials are actually ongoing in the elderly. The first one aims to demonstrate the efficacy of the Ad26.RSV.Pre-F vaccine in the prevention of reverse transcriptase-polymerase chain reaction and confirmed the RSV-LRTI when compared to the placebo [54]. The second study evaluates the safety and immunogenicity of the Ad26.RSV.preF and/or RSV pre-F protein combinations [55]. Regarding children, a Phase I/IIa study evaluating the safety and tolerability of two doses given one month apart of Ad26.RSV.preF in adults and RSV-seropositive toddlers ended in April 2020. The results of the study have not yet been published [56]. Moreover, a Phase I/IIa trial started in January 2019 is evaluating whether an intramuscular regimen of thee doses of Ad26.RSV.preF is safe, well tolerated and immunogenic in RSV-seronegative toddlers [57]. ChAd155-RSV, developed by GlaxoSmithKline, is another Adenovirus-based vaccine currently in clinical development. This vaccine is intended for children and uses a viral vector chimpanzee-Adenovirus-155, encoding RSV F, N and M2-1 proteins [58]. In adults previously naturally exposed to RSV, ChAd155-RSV delivered intramuscularly was found to be well tolerated and to elicit specific humoral and cellular immune responses [58]. Phase II clinical trials in seropositive toddlers aged 12 to 23 months [59], and in likely seronegative infants aged 6 and 7 months [60], are still ongoing. Two vector based-vaccines use the parainfluenza virus (PIV) to display RSV antigens, with the aim to induce immunity against both viruses: MEDI-534, developed by MedImmune, and the recombinant Sendai virus vectored RSV (SeVRSV), developed by the National Institute of Allergy and Infectious Diseases (NIAID). MEDI-534 is based on a modified bovine PIV3, expressing the human PIV3 fusion, the human PIV3 hemagglutinin-neuraminidase and the RSV F proteins [61]. In a Phase I trial enrolling seropositive children, this live attenuated intranasal vaccine was safe but minimally immunogenic [62]. When tested in seronegative infants, who are the target population for this vaccine, MEDI-534 was well tolerated and induced an immune response against RSV in about 50% of the subjects and against hPIV3 in all cases [63]. The low immunogenicity against RSV is probably due to a decreased expression of the virus secondary to genetic changes involving the F protein [64]. Therefore, further studies are warranted to reach the genetic stabilization of MEDI-534 and increase the RSV immune response. The other PIV-based vaccine is the SeVRSV vaccine. SeVRSV is a replication-competent Sendai virus, a murine PIV1 strain, that carries the RSV F gene produced by reverse genetics technology. This vaccine has been reported to induce a humoral response and protect the lower respiratory tract from RSV in African green monkeys [65]. A Phase I trial to evaluate the safety and immunogenicity of intranasal SeVRSV in healthy adults has been completed in 2019 [66]. The results have not yet been published.

Vector-based vaccines are potentially good candidates for the paediatric population, because there is no risk of reversion to wild-type RSV and of ERD. These vaccines are still in an initial stage concerning their evaluation. The eight vaccines that have been tested in clinical trials are based on MVA, Adenovirus, bovine PIV or Sendai Virus vectors. The MVA-BN-RSV vaccine is the only one that has passed Phase 2 clinical trial, and a Phase 3 trial in the elderly has been planned for initiation in 2021.

## 4. Subunit Vaccines

Subunit vaccines are created with RSV protein fragments. They are poorly immunogenic due to their non-replicating nature and their limited components; therefore, booster administration and adjuvants are often necessary to make them effective [70]. This type of vaccine primarily induce a CD4 T cell response [71], with higher risk for vaccine-ERD in seronegative infants [20,21]. The adjuvants’ activity is essential for the creation of neutralizing antibodies and a protective response, thanks to Toll-like receptors stimulation and B-cell affinity maturation, which also help prevent ERD [70]. Nowadays, subunit vaccines are under development only for pregnant women and older people that have already had a previous RSV infection and that are not a risk of developing ERD [14] (Table 4). The most appropriate protein to design subunit vaccines is the F protein, mostly in prefusion conformation [71]. The postfusion conformation lacks important antigenic sites [72] and this could be the reason for the failure of some candidate vaccines [73]. NIAID is developing VRC-RSVRGP084-00-VP, a prefusion F-based RSV vaccine. The protein contained in this vaccine candidate is DS-Cav1, a secreted variant of F glycoprotein that has been stabilized in the prefusion native conformation thanks to protein engineering [74]. The Phase I trial of VRC 317 has recently been concluded (April 2020), and the results are expected to be published soon. During the trial, this vaccine has been tested in healthy adults with or without an aluminium adjuvant, with the aim of monitoring safety, tolerability and immunogenicity [75]. Preliminary results that were published in 2019 are promising: DS-Cav1 is able to increase the RSV neutralizing activity in serum from 7- to 15-fold, increasing the pre-F-specific antibodies [76]. Another vaccine candidate based on the prefusion F protein is being developed by Pfizer. There are two ongoing Phase I/II trials (with and without adjuvant) who enlist healthy adults of various age groups. Participants from certain arms of the studies will concomitantly receive a seasonal inactivated influenza vaccine. These trials will be completed at the end of 2020 and in 2021 [71,77,78]. Meanwhile, the Phase IIb trial, concluded in December 2019, recruited a group of healthy non-pregnant women (18–49 years old) to which the Pfizer’s vaccine candidate was given together with diphtheria/tetanus/pertussis vaccine [79]. There is a last Pfizer’s trial in progress: a Phase IIb study currently in progress will evaluate the safety, tolerability and immunogenicity of the stabilized prefusion F subunit vaccine in pregnant participants and will assess the safety and characteristics of the transplacentally transferred antibodies in their infants [80]. DepoVax (DPX)-RSV is a candidate based on the ectodomain of the SH protein (She), presented with a novel lipid-based formulation (DepoVax) that ensures a prolonged exposure of the antigen. Phase I results have shown a good safety profile and a sustained serum IgG response (lasting more than a year) [81]. Phase II has not yet started [82]. G glycoprotein was also used to create a subunit vaccine candidate, BARS13: the Phase I study has involved healthy adult volunteers, but the results have not yet been published [82,83]. Unfortunately, the vaccine called GSK3003891A, after two Phase I trials and three Phase II studies that have established its safety and immunogenicity, is no longer under development because of the instability of the pre-F antigen during manufacturing [14,71,82]. However, GSK is working on two new candidates, in either a prefusion or an undisclosed conformation: GSK3888550A, with pregnant women as the target population, and GSK3844766A, designed for older adults [82]. With regard to the first vaccine, only a Phase I trial has been concluded, but the results have not yet been published; a Phase II trial is assessing the safety and immune response in healthy pregnant women and in infants born to vaccinated mothers, and will be concluded in 2021 [84,85,86]. MEDI-7510, a subunit vaccine developed by MedImmune, has been proven to be immunogenic but it did not protect older adults from RSV illness in a Phase IIb trial, so studies to develop it are no longer in progress [87].

Subunit vaccines can be considered a safe choice: they do not contain live viruses that could return to a virulent state and they cannot induce an exceeding immune response. However, owing to concerns of ERD associated with protein-based vaccines, they are potentially good candidates only for pregnant women and the elderly. Observing the concluded trials and the studies in progress, induction of protective immunity and obtaining F protein conformation stability remain the major unsolved problems for creating an effective subunit vaccine.

## 5. Live-Attenuated and Chimeric Vaccines

Live-attenuated RSV vaccines (RSV-LAVs) are produced with versions of RSV that are able to replicate but have been modified to discourage severe disease. They can be created by traditional techniques (i.e., temperature or chemical sensitivity) or, thanks to an improved understanding of the RSV viral genome, by reverse genetics to create an attenuated replication-competent vaccine [91]. ERD has not been observed with RSV-LAVs or replicating vaccine vectors. For this reason, these candidates can be considered safe for naive-RSV infants [20,92]. Furthermore, RSV-LAVs have other benefits: the ability to replicate in the respiratory tract despite the presence of maternal antibodies, the capacity to promote both a humoral and cellular immune response and the possibility to be administered as nasal drops, which are less invasive and better tolerated in children [93,94] (Table 5). Most vaccines are in Phase I and no candidates of this type have progressed beyond Phase II clinical trials. Anyway, the replicative nature of RSV-LAVs and their major safety compared to other types of vaccine make them an attractive strategy for seronegative infants. A promising strategy involves deletion of the RNA synthesis regulatory protein M2-2, resulting in increased viral RNA gene transcription and antigen expression but decreased genome replication. Two candidate vaccines, MEDI/ΔM2-2 and LID/ΔM2-2, have originally been evaluated. Both induced strong RSV-neutralizing antibody responses; however, the LID/ΔM2-2 vaccine has been considered more effective, as it confers a slight increase in replication [95,96]. Although LID/ΔM2-2 was well tolerated in the Phase I study, the higher replication might make it poorly tolerated when administrated to a larger population. Other candidates containing the M2-2 protein mutations have been tested. LID/cp/ΔM2-2, designed with the insertion of a set of five defined point mutations originally derived from serial cold passage, resulted in an over-attenuated vaccine that had low infectivity and low-titre antibodies in only a fraction of the participants, thus being not suitable for further development [97]. Another vaccine constructed by the addition of the genetically stabilized mutation 1030s, LID/ΔM2-2/1030s, was investigated in a Phase I study [98]. This candidate was more attenuated than the parent vaccine with vaccine-induced titres of serum RSV-neutralizing antibodies, essentially equivalent to a primary RSV infection [98]. These characteristics make this candidate very attractive for further investigations. D46/NS2/N/ΔM2-2-HindIII was built on the LID backbone but with several modifications that are expected to generate a phenotype similar to MEDI/ΔM2-2. This vaccine has been demonstrated to have a greater attenuation than LID/ΔM2-2, but considerably higher peak viral titres than MEDI/ΔM2-2 [99]. The pre-fusion RSV protein is a structural protein that plays an important role as a natural immunogen. Stobart et al. identified a chimeric RSV strain with enhanced pre-fusion antigen levels, thermostability and immunogenicity, despite heavy attenuation in the airways of cotton rats generating a promising RSV-LAV candidate [100]. The results of this Phase I clinical trial have not yet been published. MEDI-559, a candidate that contains five attenuating mutations involving the nucleoprotein and fusion, the large polymerase M2-1 and M2-2 proteins, was a promising candidate. However, after administration, it appeared to be genetically unstable, leading to a partial loss of its phenotype: it was more biologically active and immunogenic in vitro than in vivo, and showed a higher risk of LRTI [101]. RSVcps2 represents the stabilized version of the MEDI-559 vaccine. In a Phase I clinical trial, RSVcps2 was well tolerated and moderately immunogenic [102]. Another promising strategy involves deletion of the non-structural (NS) proteins 1 and 2, which modulate the immune response to promote transforming growth factor-β-mediated cell cycle arrest and viral replication [103]. In fact, it has been seen that deletion of the NS2 gene diminished RSV replication in chimpanzees [104]. The increased interferon response to infection may enhance the adaptive immune response, as has been demonstrated in calves [105]. NS2 also functions as a pathogenicity factor, promoting epithelial cell shedding in vitro and in a hamster model, potentially contributing to small airway disease [106]. Karron et al. conducted a stepwise Phase I evaluation of RSV/ΔNS2/Δ1313/I1314L, demonstrating that it is attenuated yet immunogenic in RSV seronegative children [107]. Other RSV-LAVs that include NS2 or NS1 gene modification, such as RSV 6120/∆NS2/1030s and RSV 6120/ΔNS1, are under investigation. The results have not yet been published [108,109,110,111].

LAV candidates are considered safe for clinical evaluation in children because these vaccines are not expected to cause ERD. RSV-LAVs are able to generate a robust immune response and, thanks to their safety and nature, can be administered also to infants. Another advantage is that this type of vaccine is administered intranasally, avoiding the use of needles. Even if most clinical trials are only in Phase I, RSV-LAVs represent promising candidates worth of further investigation.

The only chimeric vaccine candidate is rBCG-N-hRSV [112]. It consists of the bacillus Calmette-Guerin (BCG) vaccine expressing the nucleoprotein (N) of RSV. It is delivered via a BCG strain and induces a Th1 and a humoral response. It is the only LAV that combines protection against two respiratory pathogens, *Mycobacterium tuberculosis* and RSV. Furthermore, it could be safe for administration to newborn babies. Céspedes et al. demonstrated that this vaccine is safe, showing no side effects in mice. A Phase I clinical trial on adults has been conducted, even if the results have not been reported yet [113].

## 6. Monoclonal Antibodies

In parallel with the development of a vaccine, passive immunoprophylaxis has also been studied as an alternative approach to RSV prevention. Before the year 2000, studies had already shown that intravenous immunoglobulins against RSV prevent severe RSV respiratory disease, and in 1996, RespiGam (Massachusetts Public Health Biologic Laboratories, and MedImmune) was approved by the FDA for use in high-risk groups [116] (Table 6). The new millennium has seen the development of humanized mAb directed against the F and G surface glycoproteins encoded by RSV. The first and most studied mAb is palivizumab (MEDI-493, Synagis, MedImmune, Inc., Gaithersburg, MD). It is a humanized IgG1 mAb that binds to the F-RSV surface glycoprotein and is currently the only mAb licensed for the RSV infection in infants and children, whose indications are now restricted to premature birth, chronic lung disease and hemodynamically significant congenital heart disease [17]. Despite its great effectiveness, palivizumab has important limitations, such as needing to be administered by intramuscular injection once a month during the RSV season and a high cost, which make this approach unfeasible for healthy infants. Consequently, more recent research has pointed to the development of antibodies with a higher potency or with an extended serum half-life [116,117]. A first attempt has been done with motavizumab (MEDI-524, MedImmune). This is an RSV-specific mAb developed by remodelling the heavy and light chains of palivizumab, so that an approximately 70-fold higher affinity for the F protein of RSV and 20-fold higher potency than palivizumab were obtained [118]. Many trials have been conducted in infants, comparing motavizumab to the placebo or to palivizumab, and they have shown that motavizumab has a pharmacokinetic profile similar to palivizumab [118]. Given the likelihood that both products could be available concurrently for commercial use, and that children could receive both agents during the same RSV season, a specific trial was conducted in which infants received, sequentially, motavizumab and palivizumab, demonstrating similar results in terms of efficacy and safety [119]. Motavizumab showed comparable results to palivizumab also when tested in 1236 children with hemodynamically significant congenital heart disease [120]. Nevertheless, in a very large Phase III, randomized, double-blind, palivizumab-controlled study enrolling a total of 6635 children, a large number of cutaneous adverse events was reported in the motavizumab cohort, and the FDA decided not to approve its license [116,121]. Eventually, the effect of motavizumab on RSV viral load has also been studied, with an apparent antiviral activity that could not be confirmed in subsequent trials [122,123]. Another tested product is suptavumab, also called REGN2222, an IgG1 monoclonal antibody directed against the RSV F glycoprotein, conceived for prophylaxis in infants who do not meet the criteria for palivizumab use. In 2015, a first-in-human study, on 132 healthy adults who received intravenously or intramuscularly REGN2222 compared to the placebo, demonstrated that suptavumab was well tolerated at all evaluated doses without serious adverse events or dose-limiting toxicities [124]. More recently, a Phase III trial enrolling preterm infants (<35 weeks and 6 days of gestational age) with less than 6 months of age was conducted. In August 2017, the Sponsor Agency announced that suptavumab did not meet its primary endpoint of preventing medically attended RSV infections in infants, so this mAb was discontinued [125]. A new possibility is represented by MEDI8897, also known as nirsevimab, a mAb that targets antigenic site ∅ on the pre-F conformation of RSV, and whose half-life has been increased three-fold with a triple amino acid substitutions, reaching the possibility of protection against RSV for an entire season with a single intramuscular injection [117]. Many randomized, double-blind, placebo-controlled clinical trials, conducted first in healthy adults and then in healthy preterm infants, have shown a favourable safety profile and a mean half-life of about 80–120 days for nirsevimab, so that it could be proposed as a once-per-RSV-season prophylactic agent [126,127]. In another trial, comparison between 969 infants who received nirsevimab and 484 infants who received placebo revealed a significantly lower RSV-LRTI incidence and hospitalization rate for MEDI-8897 vs. the placebo [19]. Given these promising data, a Phase III study is currently enrolling healthy late preterm and term infants, to determine the efficacy for MEDI-8897 in this population who would not be eligible to receive RSV prophylaxis [128]. Moreover, in July 2019, a Phase II/III randomized, double-blind, palivizumab-controlled study was started to evaluate the safety of MEDI-8897 compared to palivizumab in high-risk children [129]. Finally, nirsevimab will be tested in a Phase II, open-label, uncontrolled, single-dose study to evaluate its efficacy in immunocompromised Japanese children aged ≤2 years [130]. Lastly, MK-1654 is another studied mAb, binding to site IV of the F glycoprotein, with an extended half-life. A first double-blind, Phase I study involving 152 healthy adults has shown an apparent half-life of 70–85 days with a safety profile similar to the placebo [131,132]. Subsequently, a new trial was conducted, enrolling 80 healthy adults to determine if a single intravenous dose of MK-1654 might decrease the viral RSV load compared to the placebo. This Phase IIa double-blind, randomized, placebo-controlled study ended in March 2020, and the results are not available yet [133]. Another double-blind, randomized, placebo-controlled, single ascending dose study enrolling 180 healthy pre-term or full-term (29 weeks of gestational age or higher) infants receiving one of four dose levels of MK-1654 intramuscularly started in September 2018 and it is still recruiting [134].

The use of passive prophylaxis is an alternative to active vaccination. The success of palivizumab and the challenges related to the development of an effective RSV vaccine have spurred new research in this field. However, the high cost of passive prophylaxis is still limiting its use and calls for the development of a cost-effective vaccine.

## 7. Conclusions

Development of an effective vaccine to protect high-risk groups from severe RSV infections is of critical importance, but still challenging. Different antigens and vaccine formats should be considered for different target populations (children, the elderly and pregnant women). To date, the formats that are being evaluated are particle-based vaccines, vector-based vaccines, subunit vaccines and LAVs. Currently, the only vaccine that has reached Phase III clinical trials is a maternal RSV F nanoparticle vaccine, which showed efficacy in reducing hospitalization and RSV-LRTI in young infants, but did not meet the desired primary outcome, so future trials are needed to confirm its efficacy. The same vaccine with adjuvants was safe and immunogenic in the elderly, and future clinical trials will evaluate its efficacy. Only one vector-based vaccine has passed a Phase II clinical trial, and a Phase III trial in the elderly has been planned for initiation in 2021. Subunit vaccines are potentially good candidates for pregnant women and the elderly; induction of protective immunity and obtaining F protein conformation stability remain the major unsolved problems for creating an effective subunit vaccine. Most clinical trials for RSV-LAVs are only in Phase I, but these vaccines represent promising candidates worthy of further investigation as they are able to generate a robust immune response and, thanks to their safety and nature, can be administered also to infants. While waiting for commercially available safe and effective vaccines, immune prophylaxis in selected groups of high-risk populations is still mandatory. Monoclonal antibodies with a better cost-effectiveness ratio than palivizumab, such as nirsevimab or MK-1654, are the subjects of clinical trials. In young infants, combining the use of passive immunization via maternal vaccination or mAbs, followed by paediatric active immunization, may be effective to prevent severe RSV infection. Research on this topic is still of utmost importance.

## Figures and Tables

**Table 1 vaccines-08-00672-t001:** Summary of the RSV vaccines and monoclonal antibodies in clinical development by target population. Only the most advanced trial for a specific target group is reported. RSV: respiratory syncytial virus; NIAID: National Institute of Allergy and Infectious Diseases; mAb: monoclonal antibody.

Target Population	Vaccine Name	Sponsor	Vaccine Type	Clinical Trial Phase
Pregnant women	ResVax	Novavax	Particle-based	III
	RSV F DS-Cav1 VRC-RSVRGP084-00-VP	NIAID	Subunit	I
	RSV vaccine	Pfizer	Subunit	II
	GSK3888550A	GlaxoSmithKline	Subunit	II
Children	RSV F nanoparticle	Novavax	Particle-based	I
	SynGEM	Mucosis	Particle-based	II
	Ad26.RSV.preF	Janssen	Vector-based	I/IIa
	ChAd155-RSV	GlaxoSmithKline	Vector-based	II
	MEDI-534	MedImmune (AstraZeneca)	Vector-based	I
	SeVRSV	NIAID	Vector-based	I
	RSV MEDI ΔM2-2	NIAID	Live-attenuated	I/II
	RSV LID ΔM2-2	NIAID	Live-attenuated	I
	RSV cps2	NIAID	Live-attenuated	I
	RSV LID cp ΔM2-2	NIAID	Live-attenuated	I
	RSV LID ΔM2-2 1030s	NIAID	Live-attenuated	I
	RSV D46/NS2/N/ΔM2-2-HindIII	NIAID	Live-attenuated	I
	RSV ΔNS2 Δ1313 I1314L	NIAID	Live-attenuated	I
	RSV 6120/ΔNS1; RSV 6120/F1/G2/ΔNS1	NIAID	Live-attenuated	I
	RSV ΔNS2/Δ1313/I1314L; RSV 6120/ΔNS2/1030s; RSV 276	NIAID	Live-attenuated	I/II
	RSV 6120/∆NS2/1030s	NIAID	Live-attenuated	I
	RSV ΔNS2/Δ1313/I1314L; RSV 276	NIAID	Live-attenuated	I
	RSV D46 cpΔM2-2	NIAID	Live-attenuated	I
	MEDI-559	MedImmune (AstraZeneca)	Live-attenuated	I/II
	MV-012-968	Meissa Vaccines	Live-attenuated	I
	rBCG-N-hRSV	Pontificia Universidad Catolica de Chile	Chimeric	I
	MEDI-8897 (nirsevimab)	MedImmune (AstraZeneca)	mAb	III
	MK-1654	Merck Sharp and Dohme Corporation	mAb	IIa
Elderly	RSV F nanoparticle	Novavax	Particle-based	III
	SynGEM	Mucosis	Particle-based	II
	MVA-BN RSV	Bavarian Nordic	Vector-based	II
	VXA-RSVf oral	Vaxart	Vector-based	I
	Ad26.RSV.preF	Janssen	Vector-based	II
	PanAd3-RSV and MVA-RSV	ReiThera	Vector-based	I
	DPX-RSV	Dalhousie University	Subunit	I
	RSV vaccine	Pfizer	Subunit	II
	GSK3844766A	GlaxoSmithKline	Subunit	I

**Table 2 vaccines-08-00672-t002:** Overview of the RSV particle-based vaccines under clinical development. Ab: antibody; ADA: anti-drug antibody; AE: adverse event; Ag: antigen; CI: confidence interval; CHD: congenital heart defect, CLD: chronic lung disease; COPD: chronic obstructive pulmonary disease; Corp: corporation; f.up: follow-up; GA: gestational age; IM: intramuscular; IN: intranasal; IV: intravenous; LRTD: lower respiratory tract disease; LRTI: lower respiratory tract infection; mAb: monoclonal antibody; NA: not available; NIAID: National Institute of Allergy and Infectious Diseases; PCA: palivizumab-competing antibody; PIV3: Parainfluenza Virus type 3; Ref: reference; RSV: respiratory Syncytial Virus; RT-PCR: reverse transcriptase polymerase chain reaction; SAE: serious adverse event; TCID_50_: Tissue Culture Infectious Dose 50%; vs.: versus.

Vaccine Name, Sponsor	Antigen or Target Site	Phase, Ref.	Study Population	Route	Study ID	Enrolment Time and Cohort	Summary of Published Results
ResVax, Novavax	Recombinant F protein exhibiting post-F morphology, with or without an aluminium phosphate adjuvant	I [40]	Women of childbearing age	IM	NCT01290419	December 2010–December 2011 (*n* = 150)	- Well tolerated, no dose-related increases in SAEs; - Anti-RSV F IgG titres and PCA induced at levels that is reported to be associated with decreased risk of hospitalization
		II [27]	Women of childbearing age	IM	NCT01704365	October 2012–May 2013 (*n* = 330)	- Well tolerated and immunogenic; - Most robust Ab response in 2-doses (60 or 90 µg) adjuvanted regimens; - Reduced RSV infection in recipients (11% vs. 21%, *p* = 0.04)
		II [28]	Women of childbearing age	IM	NCT01960686	October 2013–April 2014 (*n* = 720)	- Well tolerated and immunogenic; - Most robust Ab response with 120 μg and 0.4 mg aluminium formulation; - RSV infection was reduced by 52% in vaccines (*p* = 0.009)
	Recombinant F protein exhibiting post-F morphology, with an aluminium phosphate adjuvant	II [31]	Third-trimester pregnant women	IM	NCT02247726	September 2014–July 2016 (*n* = 50)	- Well tolerated and immunogenic; - Transplacental antibody transfer was 90–120% across assays for infants of vaccinated women and it was higher in women with an interval of more than 30 days between vaccination and delivery
		III [32]	Pregnant women (28–36 weeks of GA)	IM	NCT02624947	December 2015–June 2020 (*n* = 4636)	Vaccine efficacy: - 39.4% in reducing RSV-specific medically significant LRTI in young infants (<3 months of age); - 58.8% efficacy in reducing RSV-related severe hypoxemia in young infants; - 44% efficacy in reducing RSV-LRTI hospitalization
RSV F nanoparticle, Novavax	Recombinant F protein exhibiting post-F morphology +/− an aluminium phosphate adjuvant	I [29]	Elderly	IM	NCT01709019	October 2012–March 2014 (*n* = 220)	- Acceptable safety profile, no vaccine-related AEs; - Most robust immune response in adjuvanted formulations compared to high dose (90 μg vs. 60 μg RSV F protein)
	Recombinant F protein exhibiting post-F morphology without an adjuvant	II [41]	Elderly	IM	NCT02266628	October 2014–March 2016 (*n* = 1599)	- Results NA yet
		II [42]	Elderly	IM	NCT02593071	October 2015–November 2016 (*n* = 1330)	- Results NA yet
		III [34]	Elderly	IM	NCT02608502	November 2015–December 2016 (*n* = 11850)	One dose of vaccine (135 µg RSV F protein) efficacy: - No efficacy vs. RSV-symptomatic respiratory diseases and RSV-positive moderate-severe LRTI; - 61% reduction in hospitalizations due to COPD exacerbations
	Recombinant F protein exhibiting post-F morphology, adjuvant: aluminium phosphate/Matrix M	II [35]	Elderly	IM	NCT03026348	January 2017–July 2018 (*n* = 1329)	- Both adjuvants increased the magnitude, duration and quality of the immune response versus non-adjuvanted RSV F vaccine
	Recombinant F protein exhibiting post-F morphology, with or without an aluminium phosphate adjuvant	I [30]	Children (2–6 years old)	IM	NCT02296463	November 2014–April 2016 (*n* = 32)	- Well tolerated; - Anti-F IgG and PCA titres increase day 14, peak day 28, elevated to day 56; - 10-times increase PCA and anti-F IgG titres in adjuvanted group; 6-times increase in unadjuvanted group
SynGEM, Mucosis	Prefusion F	I [37]	Healthy adults	IN	NCT02958540	Unknown (*n* = 48)	- Well tolerated; - PCA titres achieved plateau after the first dose in high-dose (250 μg) recipients; after a boost at day 28 in low-dose (140 μg) recipients; - F protein site ∅-specific Ab were not detected

**Table 3 vaccines-08-00672-t003:** Overview of the RSV vector-based vaccines in clinical development. Ab: antibody; ADA: anti-drug antibody; AE: adverse event; Ag: antigen; CI: confidence interval; CHD: congenital heart defect, CLD: chronic lung disease; COPD: chronic obstructive pulmonary disease; Corp: corporation; f.up: follow-up; GA: gestational age; IM: intramuscular; IN: intranasal; IV: intravenous; LRTD: lower respiratory tract disease; LRTI: lower respiratory tract infection; mAb: monoclonal antibody; NA: not available; NIAID: National Institute of Allergy and Infectious Diseases; PCA: palivizumab-competing antibody; PIV3: Parainfluenza Virus type 3; Ref: reference; RSV: respiratory syncytial virus; RT-PCR: reverse transcriptase polymerase chain reaction; SAE: serious adverse event; TCID_50_: Tissue Culture Infectious Dose 50%; vs.: versus.

Vaccine Name, Sponsor	Antigen or Target Site	Phase, Ref.	Study Population	Route	Study ID	Enrolment Time and Cohort	Summary of Published Results
MVA-BN RSV, BavarianNordic	F, G (A and B subtypes), N, M2	I [45]	Healthy adults and older adults (50–65 years)	IM	NCT02419391	August 2015–May 2016 (*n* = 63)	- Well tolerated and immunogenic; - Higher and broader cellular and humoral immune responses in the high dose group (1 × 10^8^ TCID_50_) compared to low dose group (1 × 10^7^ TCID_50_)
		II [46]	Older adults (≥55 years)	IM	NCT02873286	September 2016–August 2018 (*n* = 420)	- Well tolerated with no vaccine-related SAEs; - A single vaccination with the high dose (1 × 10^8^ TCID_50_) induced a broad Ab (neutralizing Ab and total IgG and IgA titres) and Th1-biased cellular immune response against all 5 RSV antigens; - The immune response persisted at least 6 months and can be boosted at 12 months.
PanAd3-RSV and MVA-RSV, ReiThera	F, N and M2	I [48,49]	Healthy adults and older adults (60–75 years)	IM (PadAd3 and MVA); IN (PanAd3)	NCT01805921	March 2013–August 2015 (*n* = 72: 42 healthy adults + 30 older adults)	- Well tolerated and immunogenic; - RSV neutralising Ab titres increased after IM prime with PanAd3-RSV and after IM boost for vaccinees primed by the IN route; - Anti-F IgG and IgA Ab secreting cells increased after IM PanAd3-RSV prime and after IM MVA-RSV boost; - PanAd3-RSV prime/MVA-RSV boost induced robust RSV specific T-cell responses independent of the route of priming in adults; - RSV-specific T-cell immune responses was most marked in older adults following IM prime.
VXA-RSVf, Vaxart	F, dsDNA activating TLR3 receptor	I [51]	Healthy adults	ORAL	NCT02830932	June 2016–September 2017 (*n* = 66)	- Results NA
Ad26.RSV.Pre-F, Janssen	Prefusion F	I [67]	Healthy elderly (≥60 years)	IM	NCT02926430	November 2016–January 2019 (*n* = 73)	- Results NA
		II [53]	Healthy elderly (≥60 years)	IM	NCT03339713	December 2017–July 2018 (*n* = 180)	- The co-administration of Ad26.RSV.preF with a seasonal influenza vaccine was well tolerated and showed no evidence of interference in immune response
		II [54]	Healthy elderly (≥65 years)	IM	NCT03982199	August 2019–In progress (estimated *n* = 6672)	- Results NA
		I/IIa [56]	Adults and RSV-seropositive children (12–24 months)	IM	NCT03303625	November 2017–April 2020 (*n* = 48)	- Results NA
		I/IIa [57]	RSV-seronegative children (12–24 months)	IM	NCT03606512	January 2019–In progress (estimated *n* = 38)	- Results NA
		I [68]	Healthy adults, older adults	IM	NCT03795441	January 2019–July 2019 (*n* = 24)	- Results NA
		II [69]	Healthy adults	IM	NCT03334695	October 2016–November 2018 (*n* = 64)	- Results NA
Ad26.RSV.Pre-F +/− prefusion F, Jannsen	Purified Prefusion F	II [55]	Healthy elderly (≥60 years)	IM	NCT03502707	July 2018–In progress (estimated *n* = 669)	- Results NA
ChAd155-RSV, GlaxoSmithKline	F, N, M2-1	I [58]	RSV-seropositive adults	IM	NCT02491463	July 2015–February 2017 (*n* = 73)	- Well tolerated and safe; - ChAD155-RSV induced specific humoral and cellular immune response
		II [59]	RSV-seropositive infants (12–23 months)	IM	NCT02927873	January 2017–In progress (estimated *n* = 82)	- Results NA
		I [60]	Likely RSV-seronegative infants (6–7 months)	IM	NCT03636906	April 2019–In progress (estimated *n* = 201)	- Results NA
MEDI-534, MedImmune	Wild type F	I [62]	RSV/PIV3-seropositive children (1–9 years)	IN	NCT00345670	June 2006–May 2007 (*n* = 120)	- Acceptable safety profile; - Minimally immunogenic
		I [63]	RSV/PIV3-seronegative children (6–24 months) and infants (2 months)	IN	NCT00686075	June 2008–August 2012 (*n* = 1338)	- Published results on 49 healthy children (6–24 months); - Well tolerated and safe; - Seroresponse to RSV and PIV3 highest in high dose (1 × 10^6^ TCID_50_) group
SeVRSV, NIAID	Wild type F	I [66]	Healthy adults	IN	NCT03473002	May 2018–February 2019 (*n* = 21)	- Results NA

**Table 4 vaccines-08-00672-t004:** Overview of the RSV subunit vaccines in clinical development. Ab: antibody; ADA: anti-drug antibody; AE: adverse event; Ag: antigen; CI: confidence interval; CHD: congenital heart defect, CLD: chronic lung disease; COPD: chronic obstructive pulmonary disease; Corp: corporation; f.up: follow-up; GA: gestational age; IM: intramuscular; IN: intranasal; IV: intravenous; LRTD: lower respiratory tract disease; LRTI: lower respiratory tract infection; mAb: monoclonal antibody; NA: not available; NIAID: National Institute of Allergy and Infectious Diseases; PCA: palivizumab-competing antibody; PIV3: Parainfluenza Virus type 3; Ref: reference; RSV: respiratory syncytial virus; RT-PCR: reverse transcriptase polymerase chain reaction; SAE: serious adverse event; TCID_50_: Tissue Culture Infectious Dose 50%; vs.: versus.

Vaccine Name, Sponsor	Antigen or Target Site	Phase, Ref.	Study Population	Route	Study ID	Enrolment Time and Cohort	Summary of Published Results
RSV F DS-Cav1, NIAID	Prefusion stabilised trimeric F protein with or without alum adjuvant	I [75,76]	Healthy adults (18–50 years)	IM	NCT03049488	February 2017–January 2020 (*n* = 100)	Interim analysis after the first dose (50 or 150 mg, ± aluminium hydroxide—alum, *n* = 10 subjects per group): - After 4 weeks, neutralizing activity was respectively increased 7-, 12- and 15-fold increased after 50 mg and 150 mg of DS-Cav1 ± alum (all *p* < 0.001); - Neutralization remained 5- to 10-fold above baseline after 12 weeks (*p* < 0.001); - Well tolerated without any SAE reported
DPX-RSV, Dalhousie University And ImmunoVaccine Technologies	Ectodomain of the small hydrophobic glycoprotein (SHe), presented with a novel lipid-based formulation (DepoVax)	I [81,88]	Healthy adults (50–64 years)	IM	NCT02472548	May 2015–March 2017 (*n* = 40)	Two dose levels (10 or 25 μg) of SHe compared with the placebo, booster dose on day 56: - Highly immunogenic, sustained Ag-specific Ab responses up to 180 days after the second dose and up to 421 days in the higher-dose group; - Well tolerated;
RSV vaccine, Pfizer	Prefusion F protein	II [79]	Adults	IM	NCT04071158	October 2019–December 2019 (*n* = 713)	- Results NA
		II [80]	Pregnant women	IM	NCT04032093	August 2019 –In progress (estimated *n* = 650)	- Results NA
		II [78]	Healthy elderly	IM	NCT03572062	June 2018–In progress (estimated *n* = 317)	- Results NA
		II [77]	Adults	IM	NCT03529773	April 2018–In progress (estimated *n* = 1235)	- Results NA
BARS13, (Advanced Vaccine Laborat)	G protein	II [83]	Healthy adults	IM	ACTRN12618000948291	October 2018–October 2019 (*n* = 60)	- Results NA
GSK3888550A, GlaxoSmithKline	F protein	II [84]	Women of childbearing age	IM	NCT03674177	October 2018–September 2019 (*n* = 502)	- Results NA
		II [85]	Women of childbearing age	IM	NCT04138056	September 2019–August 2020 (*n* = 509)	- Results NA
		II [86]	Pregnant women	IM	NCT04126213	November 2019–In progress (estimated *n* = 410)	- Results NA
GSK3844766A, GlaxoSmithKline	F Protein	I [89]	Elderly	IM	NCT03814590	January 2019–In progress (estimated *n* = 1055)	- Results NA
		I [90]	Elderly	IM	NCT04090658	September 2019–In progress (estimated *n* = 40)	- Results NA

**Table 5 vaccines-08-00672-t005:** Overview of the RSV live-attenuated or chimeric vaccines in clinical development. Ab: antibody; ADA: anti-drug antibody; AE: adverse event; Ag: antigen; CI: confidence interval; CHD: congenital heart defect, CLD: chronic lung disease; COPD: chronic obstructive pulmonary disease; Corp: corporation; f.up: follow-up; GA: gestational age; IM: intramuscular; IN: intranasal; IV: intravenous; LRTD: lower respiratory tract disease; LRTI: lower respiratory tract infection; mAb: monoclonal antibody; NA: not available; NIAID: National Institute of Allergy and Infectious Diseases; PCA: palivizumab-competing antibody; PFU: plaque-forming units; PIV3: Parainfluenza Virus type 3; Ref: reference; RSV: respiratory syncytial virus; RT-PCR: reverse transcriptase polymerase chain reaction; SAE: serious adverse event; TCID_50_: Tissue Culture Infectious Dose 50%; vs.: versus.

Vaccine Name, Sponsor	Antigen or Target Site	Phase, Ref.	Study Population	Route	Study ID	Enrolment Time and Cohort	Summary of Published Results
RSV MEDI ΔM2-2, NIAID MedImmune (AstraZeneca)	RSV MEDI ΔM2-2	I [96]	Children and Adults	IN	NCT01459198	August 2011–August 2015 (*n* = 60)	- Median peak virus titre: 1.5 log 10 PFU/mL; - Detected Abs in 95%
RSV cps2, NIAID MedImmune (AstraZeneca)	RSV cps2	I [102]	RSV-seronegative children (6–24 months)	IN	NCT01968083 NCT01852266	Oct 2013–April 2015 (*n* = 51 + 50)	- Median peak virus titre: 2.9 log10 copies/mL; - Detected Abs in 59%
RSV LID ΔM2-2, NIAID Sanofi Pasteur	RSV LID ΔM2-2	I [95]	Children (6–24 months)	IN	NCT02040831 NCT02237209	January 2014–December 2016 (*n* = 3 + 29)	- Median peak virus titre: 3.8 log 10 PFU/mL; - Detected Abs in 90%
RSV LID ΔM2-2 1030s, NIAID Sanofi Pasteur	RSV LID ΔM2-2 1030s	I [98]	Children (6–24 months)	IN	NCT02794870 NCT02952339	July 2015–July 2017 (*n* = 33 + 33)	- Median peak virus titre: 3.1 log 10 PFU/mL; - detected Abs in 90%
RSV LID cp ΔM2-2, NIAID	RSV LID cp ΔM2-2	I [97]	Children (6–24 months)	IN	NCT02890381 NCT02948127	October 2015–April 2018 (*n* = 13 + 8)	- Median peak virus titre: 4.5 log 10 PFU/mL; - detected Abs in 56%
RSV D46/NS2/N/ΔM2-2-HindIII, NIAID	RSV D46/NS2/N/ΔM2-2-HindIII	I [99]	RSV-seronegative children (6–24 months)	IN	NCT03102034 NCT03099291	April 2017–May 2018 (*n* = 6 + 32)	- Mean peak: 6.1 log 10 copies/mL; - Detected Abs in 95%
RSV ΔNS2 Δ1313 I1314L, NIAID	RSV ΔNS2 Δ1313 I1314L	I [107]	Children - RSV-seropositive (12–59 months); - RSV-seronegative (4–24 months)	IN	NCT01893554	June 2013–October 2018 (*n* = 105)	- Mean peak virus titre: 3–3.5 log 10 PFU/mL; - Detected Abs in 90%
RSV 6120/ΔNS1; RSV 6120/F1/G2/ΔNS1, NIAID	RSV 6120/ΔNS1; RSV 6120/F1/G2/ΔNS1	I [114]	Children - RSV-seropositive (12–59 months); - RSV-seronegative (6–24 months)	IN	NCT03596801	June 2018–In progress (estimated *n* = 75)	- Results NA
RSV ΔNS2/Δ1313/I1314L; RSV 6120/ΔNS2/1030s; RSV 276, NIAID	RSV ΔNS2/Δ1313/I1314L; RSV 6120/ΔNS2/1030s; RSV 276	I/II [108]	RSV-seronegative children (6–24 months)	IN	NCT03916185	May 2019–In progress (estimated *n* = 160)	- Results NA
RSV 6120/∆NS2/1030s, NIAID	RSV 6120/∆NS2/1030s	I [109]	Children - RSV-seropositive (12–59 months); - RSV-seronegative (6–24 months)	IN	NCT03387137	October 2017–In progress (estimated *n* = 45)	- Results NA
RSV ΔNS2/Δ1313/I1314L; RSV 276, NIAID	RSV ΔNS2/Δ1313/I1314L; RSV 276	I [110]	RSV-seronegative children (6–24 months)	IN	NCT03422237	October 2017–September 2020 (*n* = 80)	- Results NA
RSV ΔNS2 Δ1313 I1314L, NIAID	RSV ΔNS2 Δ1313 I1314L	I [111]	Children (6–24 months)	IN	NCT03227029	Aug 2017–September 2020 (*n* = 80)	- Results NA
RSV D46 cpΔM2-2, NIAID	RSV D46 cpΔM2-2	I [115]	Children (6–60 months)	IN	NCT02601612	October 2015–September 2019 (*n* = 45)	- Results NA
MV-012-968, Meissa Vaccines	MV-012-968	I [100]	Adult	IN	NCT04227210	January 2020–August 2020 (*n* = 20)	- Results NA
MEDI-559, MedImmune (AstraZeneca)	MEDI-559	I/II [101]	RSV-seronegative children (6–24 months)	IN	NCT00767416	October 2008–December 2011 (*n* = 116)	
rBCG-N-hRSV, Pontificia Universidad Catolica de Chile	rBCG-N-hRSV	I [112]	Adults	IN	NCT03213405	June 2017–May 2018 (*n* = 24)	- Results NA

**Table 6 vaccines-08-00672-t006:** Overview of the anti-RSV monoclonal antibodies in clinical development. Ab: antibody; ADA: anti-drug antibody; AE: adverse event; Ag: antigen; CI: confidence interval; CHD: congenital heart defect, CLD: chronic lung disease; COPD: chronic obstructive pulmonary disease; Corp: corporation; f.up: follow-up; GA: gestational age; IM: intramuscular; IN: intranasal; IV: intravenous; LRTD: lower respiratory tract disease; LRTI: lower respiratory tract infection; mAb: monoclonal antibody; NA: not available; NIAID: National Institute of Allergy and Infectious Diseases; PCA: palivizumab-competing antibody; PIV3: Parainfluenza Virus type 3; Ref: reference; RSV: respiratory syncytial virus; RT-PCR: reverse transcriptase polymerase chain reaction; SAE: serious adverse event; TCID_50_: Tissue Culture Infectious Dose 50%; vs.: versus.

Vaccine Name, Sponsor	Antigen or Target Site	Phase, Ref.	Study Population	Route	Study ID	Enrolment Time and Cohort	Summary of Published Results
Nirsevimab, (MEDI-8897), MedImmune (AstraZeneca)	Antibody targeting site Ø of the F protein of RSV (extended half-life obtained with YTE mutation in Fc)	I [126]	Healthy adults	IV or IM	NCT02114268	April 2014–June 2015 (*n* = 136)	Single dose of MEDI-8897 (*n* = 102) in 1 of 5 cohorts (300, 1000, 3000 mg iv or 100, 300 mg im) vs. placebo (*n* = 34), f.up for 360 days - Mean half-life 85–117 days; - similar ADA responses (mAb 13.7% vs. placebo 15.2%)
		Ib/IIa [127]	Healthy preterm infants	IM	NCT02290340	January 2015–September 2016 (*n* = 151)	Published results on 89 healthy preterm infants, single dose of MEDI-8897 (*n* = 71) in 1 of 3 cohorts (10, 25, 50 mg im) vs. placebo (*n* = 18), follow-up for 360 days, - mean half-life 62.5–72.9 days; - different ADA responses (mAb 26.5% vs. placebo 0%)
		IIb [19]	Healthy preterm infants	IM	NCT02878330	November 2016–November 2017 (*n* = 1453)	Single dose of 50 mg im MEDI-8897 (*n* = 969) vs. placebo (*n* = 484), follow-up for 360 days - significantly lower RSV-LRTI incidence for MEDI-8897 vs. placebo (relative difference 70.1% (95% CI 52.3–81.2), *p* < 0.001); - significantly lower RSV-LRTI hospitalization for MEDI-8897 vs. placebo (relative difference 78.4% (95% CI 51.9–90.3), *p* < 0.001)
		III [128]	Healthy late preterm and term infants (born at or after 35 weeks GA, aged <1 year)	NA (IM?)	NCT03979313	July 2019–In progress (estimated *n* = 3000)	- Results NA
		II/III [129]	High-risk children (preterm infants born ≤35 weeks GA without CLD/CHD, infants with CLD of prematurity, hemodynamically significant CHD)	NA (IM?)	NCT03959488	July 2019–In progress (estimated *n* = 1500)	- Results NA
		II [130]	Immunocompromised Japanese children aged < 2 years	IM	NCT04484935	Not recruiting yet	- Results NA
MK-1654, Merck Sharp and Dohme Corp.	Antibody targeting site IV of the F protein of RSV with an extended half-life	IIa [133]	Healthy adults	IV	NCT04086472	October 2019–In progress (*n* = 80)	- Results NA
		I/II [134]	Healthy pre-term (born at 29–35 weeks GA) and full-term (born at >35 weeks GA) infants	IM	NCT03524118	September 2018–In progress (estimated *n* = 180)	- Results NA

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
