# Peer review of "Current State and Challenges in Developing Respiratory Syncytial Virus Vaccines"

_vaccines, 2020, doi:10.3390/vaccines8040672_

Round 1

Reviewer 1 Report

The manuscript by Biagi et al is a comprehensive review of the current vaccine strategies for RSV. The authors list all the vaccine candidates that have progressed to clinical trials, regardless of whether the development is still continuing or not. This will be an excellent resource for any researcher entering the field of RSV vaccine development.

There are a few minor comments:

  1. Abstract. The authors use the term 'pharmacologic research' which usually refers to drugs to treat a condition. I suggest changing this to a term that more clearly demonstrates the subject matter of the review, i.e. vaccines.
  2. Abstract. I don't understand what is meant by '...targeting different subjects' (line 18-19). Do you mean different groups of patients? Or different types of vaccines? Please modify so that this sentence is easy to interpret.
  3. Abstract. There is one 'throw-away' line about the anti-RSV MAbs which is confusing. I suggest re-writing it to be closer to the statements in the 'conclusion' section.
  4. Line 51-52 in section 1.2. insertion of viral RNA does not induce syncytia as suggested by this sentence. Please re-write.
  5. There are several small typographical errors, e.g. line 38 - 17.000, should this be '17000'? Line 85 - there should be a space between the full stop and the start of the next sentence. (there are several instances of this, please go through the manuscript and correct). Line 104 should read - Since severe RSV... Line 174 - data that had. Table 2, information about SynGEM Mucosis vaccine, change 'dat' to 'day'. Line 228 ..intramuscularly was found to be well.. Line 268 - Phase I trial 'of'. Line 280 - A Phase IIb study currently in progress will... Line 314 - nasal drops, that are less.. Line 340 - have not yet been published. Line 345 - RSVcps2 was well-tolerated. Line 372 - ...RSV prevent severe RSV disease.. Line 409 - ...has been increased threefold with ... Line 419 - ...2019, a Phase II/III .... study was started ... Line 446 - ...adjuvants was safe ... Line 46 - are the subject of clinical trials.
  6. Modification of the text to remove the above and similar small text errors will greatly add to the readability of the manuscript.

Author Response

Point-by-point reply to Reviewer 1:

The manuscript by Biagi et al is a comprehensive review of the current vaccine strategies for RSV. The authors list all the vaccine candidates that have progressed to clinical trials, regardless of whether the development is still continuing or not. This will be an excellent resource for any researcher entering the field of RSV vaccine development.

Reply: We thank the reviewer for the nice comments about our paper.

There are a few minor comments:

  1. Abstract. The authors use the term 'pharmacologic research' which usually refers to drugs to treat a condition. I suggest changing this to a term that more clearly demonstrates the subject matter of the review, i.e. vaccines.

Reply: we have changed the term into “research in this field” (line 18).

  1. Abstract. I don't understand what is meant by '...targeting different subjects' (line 18-19). Do you mean different groups of patients? Or different types of vaccines? Please modify so that this sentence is easy to interpret.

Reply: we have changed “targeting different subjects” into “in different formats” (line 18)

  1. Abstract. There is one 'throw-away' line about the anti-RSV MAbs which is confusing. I suggest re-writing it to be closer to the statements in the 'conclusion' section.

Reply: thank you. We have deleted the sentence “In parallel, new monoclonal antibodies are     under evaluation to possibly extend the groups of recipients of passive immune        prophylaxis.” and added “While waiting for commercially available safe and effective   vaccines, immune prophylaxis in selected groups of high-risk populations is still     mandatory.” (lines 21-22)

  1. Line 51-52 in section 1.2. insertion of viral RNA does not induce syncytia as suggested by this sentence. Please re-write.

Reply: thank you for spotting this inaccuracy; we have rewritten the sentence (lines 57-58)

  1. There are several small typographical errors, e.g. line 38 - 17.000, should this be '17000'? Line 85 - there should be a space between the full stop and the start of the next sentence. (there are several instances of this, please go through the manuscript and correct). Line 104 should read - Since severe RSV... Line 174 - data that had. Table 2, information about SynGEM Mucosis vaccine, change 'dat' to 'day'. Line 228 ..intramuscularly was found to be well.. Line 268 - Phase I trial 'of'. Line 280 - A Phase IIb study currently in progress will... Line 314 - nasal drops, that are less.. Line 340 - have not yet been published. Line 345 - RSVcps2 was well-tolerated. Line 372 - ...RSV prevent severe RSV disease.. Line 409 - ...has been increased threefold with ... Line 419 - ...2019, a Phase II/III .... study was started ... Line 446 - ...adjuvants was safe ... Line 46 - are the subject of clinical trials.

Reply: thank you for spotting these inaccuracies. All were corrected and more were found by careful revision.

  1. Modification of the text to remove the above and similar small text errors will greatly add to the readability of the manuscript.

Reply: thank you for revising our paper and for your suggestions.

Reviewer 2 Report

This well written and comprehensive up to date review of the status of vaccines for RSV was a joy to read. It contains several complete tables and multiple references. The work described RSV vaccine history, populations requiring vaccination, and the types of vaccine candidates being tested. Further, it provides data on completed clinical trials and also brings up to date information on the status on in progress trials. 

I could not find anything wrong with this review, but must assume that the information they provide on the trials is accurate, as I am not familiar with each product that is mentioned as being in development or clinical trials. I can say that the work described appears to be very complete.

There is an additional part of the manuscript that deals with monoclonal antibody therapy/prophylaxis. This, also, is up to date and informative.

I have only one question; in line 273 the term autoantibodies is used. This is a bit confusing. Usually the term autoantibodies is used to refer to antibodies directed to self antigens, and this is, of course, not good. But the term used here seems to refer to the pre-T specific antibodies. I suspect this was a misprint. Please address as it would not be a good thing to confer the idea that the vaccine might elicit an autoimmune response.

Great Job! This will be a useful reference for many.

Author Response

Point-by-point reply to Reviewer 2

This well written and comprehensive up to date review of the status of vaccines for RSV was a joy to read. It contains several complete tables and multiple references. The work described RSV vaccine history, populations requiring vaccination, and the types of vaccine candidates being tested. Further, it provides data on completed clinical trials and also brings up to date information on the status on in progress trials. 

I could not find anything wrong with this review, but must assume that the information they provide on the trials is accurate, as I am not familiar with each product that is mentioned as being in development or clinical trials. I can say that the work described appears to be very complete.

There is an additional part of the manuscript that deals with monoclonal antibody therapy/prophylaxis. This, also, is up to date and informative.

I have only one question; in line 273 the term autoantibodies is used. This is a bit confusing. Usually the term autoantibodies is used to refer to antibodies directed to self antigens, and this is, of course, not good. But the term used here seems to refer to the pre-T specific antibodies. I suspect this was a misprint. Please address as it would not be a good thing to confer the idea that the vaccine might elicit an autoimmune response.

Great Job! This will be a useful reference for many.

Reply: thank you for your revision and your nice words. According to your suggestions, we have corrected the spotted mistake (now in line 297).

Reviewer 3 Report

This is a well-written review. I'm very pleased to read this paper and feel it is a privilege to be able to read it before it published. As a researcher worked in the respiratory viruses field, I learned new knowledge from reading your manuscript.

Here is the a couple of my comments to the manuscript.

1. While the manuscript summary each type of the vaccines with clear table.
It will be very helpful to wrote a short comments/opinions from Authors for each type of the vaccine.So the review is not just a summary, but have authors scientific thought in it.

2. It is very clear to summary each type of vaccine separately.
It will be great if the author can compare the efficacy of vaccine in each population, then we will be able to find if certain type of vaccine is better for a group of people,such as good for younger kids, or good for pregnant women, or good for the elder. Overall, I hope the author can provide move his/her own opinions after summary so many vaccines.

Thank you for the hard work to put all these progress of the RSV study together! 

Author Response

Point-by-point reply to Reviewer 3

This is a well-written review. I'm very pleased to read this paper and feel it is a privilege to be able to read it before it published. As a researcher worked in the respiratory viruses field, I learned new knowledge from reading your manuscript.

Here is the a couple of my comments to the manuscript.

  1. While the manuscript summary each type of the vaccines with clear table.
    It will be very helpful to wrote a short comments/opinions from Authors for each type of the vaccine.So the review is not just a summary, but have authors scientific thought in it.
  2. It is very clear to summary each type of vaccine separately.
    It will be great if the author can compare the efficacy of vaccine in each population, then we will be able to find if certain type of vaccine is better for a group of people,such as good for younger kids, or good for pregnant women, or good for the elder. Overall, I hope the author can provide move his/her own opinions after summary so many vaccines.

Thank you for the hard work to put all these progress of the RSV study together! 

Reply: thank you for your revision and your nice words. We have added a short paragraph at the end of each type of vaccine, where we provide a comment and identify the most suitable population for that vaccine.